# Fusion of Multispectral Aerial Imagery and Vegetation Indices for Machine Learning-Based Ground Classification

**Yanchao Zhang** [1,2,*], **Wen Yang** [1], **Ying Sun** [2], **Christine Chang** [2], **Jiya Yu** [1] **and Wenbo Zhang** [1]

1   Faculty of Machinery Engineering and Automation, Zhejiang Sci-Tech University, Hangzhou 310018, China; 201920502009@mails.zstu.edu.cn (W.Y.); 202030605314@mails.zstu.edu.cn (J.Y.); 202020602086@mails.zstu.edu.cn (W.Z.)
2   School of Integrative Plant Science, Soil and Crop Sciences Section, Cornell University, Ithaca, NY 14850, USA; ys776@cornell.edu (Y.S.); cyc54@cornell.edu (C.C.)
*   Correspondence: yczhang@zstu.edu.cn; Tel.: +86-86-571-86462181

**Abstract:** Unmanned Aerial Vehicles (UAVs) are emerging and promising platforms for carrying different types of cameras for remote sensing. The application of multispectral vegetation indices for ground cover classification has been widely adopted and has proved its reliability. However, the fusion of spectral bands and vegetation indices for machine learning-based land surface investigation has hardly been studied. In this paper, we studied the fusion of spectral bands information from UAV multispectral images and derived vegetation indices for almond plantation classification using several machine learning methods. We acquired multispectral images over an almond plantation using a UAV. First, a multispectral orthoimage was generated from the acquired multispectral images using SfM (Structure from Motion) photogrammetry methods. Eleven types of vegetation indexes were proposed based on the multispectral orthoimage. Then, 593 data points that contained multispectral bands and vegetation indexes were randomly collected and prepared for this study. After comparing six machine learning algorithms (Support Vector Machine, K-Nearest Neighbor, Linear Discrimination Analysis, Decision Tree, Random Forest, and Gradient Boosting), we selected three (SVM, KNN, and LDA) to study the fusion of multi-spectral bands information and derived vegetation index for classification. With the vegetation indexes increased, the model classification accuracy of all three selected machine learning methods gradually increased, then dropped. Our results revealed that that: (1) spectral information from multispectral images can be used for machine learning-based ground classification, and among all methods, SVM had the best performance; (2) combination of multispectral bands and vegetation indexes can improve the classification accuracy comparing to only spectral bands among all three selected methods; (3) among all VIs, NDEGE, NDVIG, and NDVGE had consistent performance in improving classification accuracies, and others may reduce the accuracy. Machine learning methods (SVM, KNN, and LDA) can be used for classifying almond plantation using multispectral orthoimages, and fusion of multispectral bands with vegetation indexes can improve machine learning-based classification accuracy if the vegetation indexes are properly selected.

**Keywords:** multispectral; vegetation indexes; information fusion; UAV; plantation classification; machine learning



## 1. Introduction

In recent years, Unmanned Aerial Vehicles (UAVs) have rapidly gained popularity as a remote sensing platform that can provide higher spatial and temporal resolution images relative to traditional media such as satellites. An increasing number of studies have utilized UAV-based near-ground remote sensing [1–4], taking advantage of their flexibility, ease of use, and ability to measure at lower altitudes in comparison with airborne sensors. The civil applications of UAV for high-resolution image acquisition have emerged as an attractive option for agriculture [5–7] and environmental monitoring [8,9]. Hence, they

were widely used in recent studies involving quantitative remote sensing applications due to their versatility. For example, different groups used hyperspectral for ground object classification and biochemical analysis [10–13], monitored ground plant diseases in an orchard or plant vegetation by using miniature thermal cameras [14–16], and used narrow-band multispectral imagery for crop water stress monitoring [17,18]. Integrating different data types to meet the diverse need of ground surveying is becoming a new research focus.

Vegetation indices have long been used in remote sensing [19–21] because they are simple, intuitionistic, and effective ways to model the ground cover reflectance. However, in drone-based remote sensing, although sensor and ground resolution are improved, vegetation indices alone cannot provide sufficient accuracy for classification [22]. Instead, new methods are needed to fully exploit the information contained within reflectance. The majority of vegetation indices were designed to enhance the difference among different land covers. For example, the normalized difference vegetation index (NDVI), the most commonly-used vegetation index in remote sensing for monitoring plant growth status, calculates the ratio of the differences and sum of the near-infrared (NIR) and red bands to represent the nonlinear structural relations between different bands. Green NDVI uses the green channel to replace red in the NDVI formula, and also represents the nonlinear relations between all multispectral channels. While many studies used machine learning methods to build models based on spectral bands for ground classification [23], the nonlinear information contained in vegetation indexes has barely been fused into ground classification models. However, the unstructured information contained within vegetation indices may help with improving classification accuracy.

Shadow is a significant factor in remote sensing. [24] took shadow reflective features into consideration during image segmentation and extracted shadows according to the statistical characteristics of the images. The authors used a set threshold to detect shade, which decreased its practicability. [25] studied the relationship between field-measured stem volume and tree crown area and tree shadow area. Shadow influences the reflectance of soil and may lead to miscalculation of some parameter retrievals, like soil organic carbon and textures [26], as well as soil moisture [27]. Moreover, shadows may lead to errors in other ground plant measurements, like tree size, crown height [28]. Therefore, a rapid and accurate shadow classification is needed for almost all remote sensing platforms, including satellite, aircraft, and UAVs.

Machine learning is widely adopted in remote sensing for ground classification, texture segmentation. Selecting features for model training input is as important as selections of samples and methods. Inclusion of a greater number of features improves the ability of the model to differentiate among species and categories. Consequently, a key goal is to find as many spatial features as possible, then use redundancy reduction or feature selection methods like Principal Component Analysis (PCA), Uninformative variables elimination (UVE), Independent Component Analysis (ICA), and lastly select different classifiers using methods such as Support Vector Machine (SVM), K-Nearest Neighbors (KNN), and Linear Discriminant Analysis (LDA) to build a statistical model for fast classification applications. Feature selection is often regarded as a process that obtains a subset from the original features set based on certain selection criteria. Usually, the dimensions of the features are reduced. Before feature selection is performed, vectors that contain normalized feature variables need to be built.

In this study, a UAV mounted with a five-band multispectral camera was used to measure an almond plantation. After the orthoimage was acquired, we first tested the classification of three surface types (tree, shadow, and soil) using different machine learning methods, namely SVM, LDA, KNN, Random forest, Decision Trees, Gradient Boost, to determine which method obtained the best performance. Secondly, we developed a fusion pipeline incorporating both spectral bands and normalized vegetation indices into the machine learning process. Thirdly, we assessed the impact of fusing spectral bands and vegetation indexes on the performance of the machine learning models. Lastly, we

comprehensively compared the results trained with vegetation indexes and determined the proper selection protocol for classification.

## 2. Materials and Methods

### 2.1. Study Area

The experiment site (36°34′42.67″N,119°26′14.01″W) is located at Reedley, CA, USA. The terrain of the experimental area is generally flat, with elevation ranging from 340 to 350 feet. A total of 20 acres of almond plantation were used for experiment. Row spacing of 2 rows was about 5.14 m, and column spacing was about 9.3 m.

### 2.2. Data Collection and Image Selection

A DJI M100 quadrotor drone was equipped with a RedEdge-M (MicaSense Inc., Washington, DC, USA) multispectral camera, and GPS (ublox AG, Swiss), and Downwelling Light Sensors (MicaSense Inc, Washington, DC, USA) for this research. The drone was powered by a 5700-mAh LiPo battery, which provides an average flight duration of ~30 min. The RedEdge-M acquires images with 1280 × 960 pixels resolution at five spectral bands including blue (475 nm), green (560 nm), red (668 nm), red edge (717 nm), and near-infrared (840 nm). The focal length of the lens was 5.4 mm and images were stored as 16-bit TIFF RAW format.

The experiment was carried out in mid-October. Multispectral images were collected at the height of 200 meters above the ground level. A high side and forward overlap rate of 80% was adopted to assure high map quality. Multispectral images were stitched together using Structure from Motion (SfM) functions in Photoscan software (Agisoft LLC, St. Petersburg, Russia) to generate an orthoimage map (Figure 1). Within the map, a region of interest was selected for analysis (Figure 1, red highlight) which contained even tree distribution and omitted signal interference from nearby road and bare soil patches.

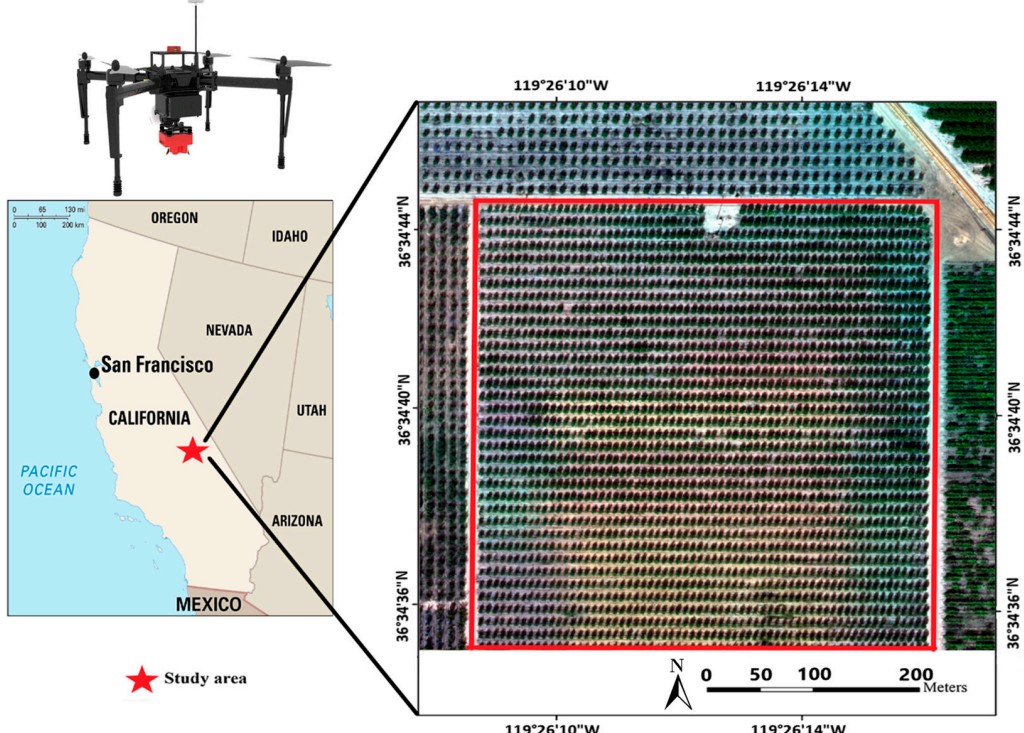

**Figure 1.** Study area: Multispectral map on 20 October, 2018 (**right**) of experiment. The region of interest used for analysis is outlined in red.

### 2.3. Selection of Spectra

As classification is based on spectral features, it is crucial to properly collect multiple spectral information in different areas to form training data sets. Sampling was performed as follows: The region of interest was split into three categories: almond tree, sunlit soil, and shade-covered soil. Circular points, consisting of a 4-pixel radius, were randomly sampled within the region of interest. The points were sampled such that the whole circle fit within a single category. The feature vector of each point was calculated as the mean value of reflectance within the circle. The whole process is as the Figure 2 shows.

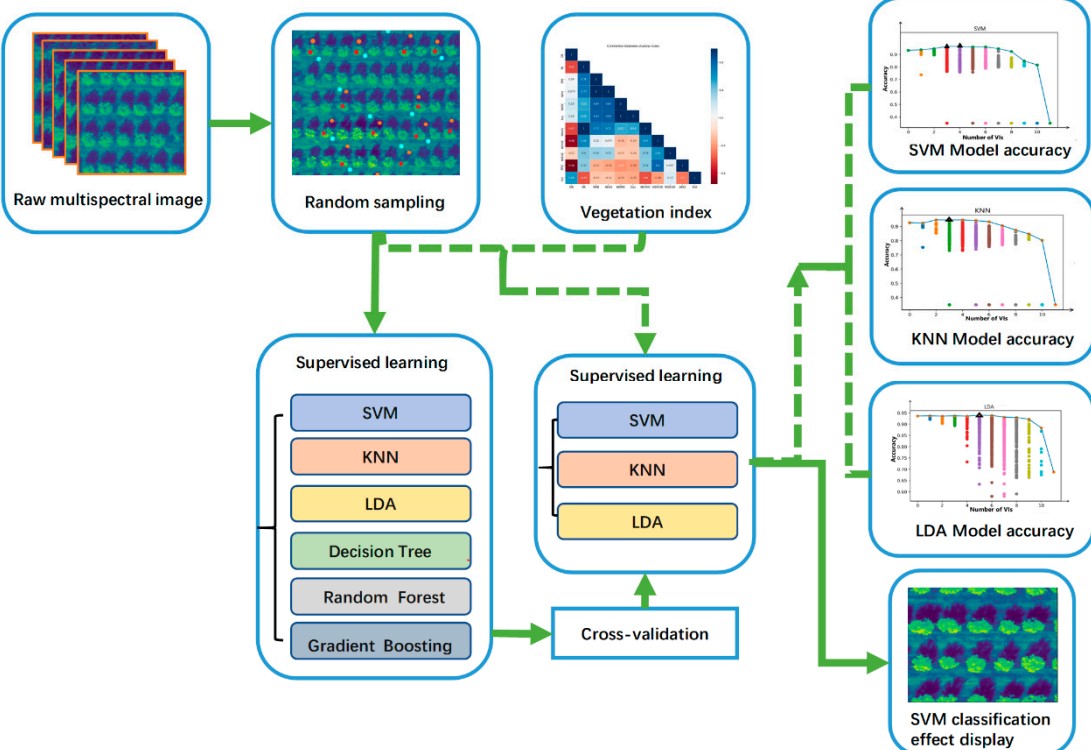

**Figure 2.** Flowchart of the data processing and image fusion.

Using Equation (1), we can calculate the actual distance of the captured object in an image, $x$, as:

$$x = h\frac{p}{f} \tag{1}$$

where $f$ represents the focal length of 5.4 mm, $p$ represents the pixel size of 0.00375 mm$^2$, and $h$ represents the height above target of 200 m. Thus, the actual distance corresponding to each pixel is 0.139 m, and the actual radius of the sampling circle is about 0.554 m.

The points' spot selection is very important in forming the dataset. If the collected areas are mostly the central area of trees, shadows, and ground, the difference between them can be ignored. However, for random selection of sample points adopted to represent the species better, it is important to collect some edge data to increase sample points' diversity. The sampling point size was designed with $4 \times 4$ pixels, which can represent the different reflectance of the species. The sampling circles where the tree is located were randomly selected. Each circle sampling area consisted of $4 \times 4$ band reflections. Inspired by the idea of averaging, the overall band reflection of it can be represented by the band reflection mean value. At the same time, the influence of noise can be avoided. Then, we labeled these 593 sample data tags as 'tree.' In the same way, we sampled the reflectance of the ground and shadows.

As shown in Table 1, we randomly extracted 400 data as a training set, 80 data as a validation set, and 113 data as the test set. Figure 3 is an orthoimage representing about 1/4 of the entire study area, enlarged to demonstrate how the sampling was performed.

**Table 1.** Data distribution of the dataset.

| Data Set | Num |
|:---:|:---:|
| Training set | 400 |
| Validation Set | 80 |
| Test Set | 113 |

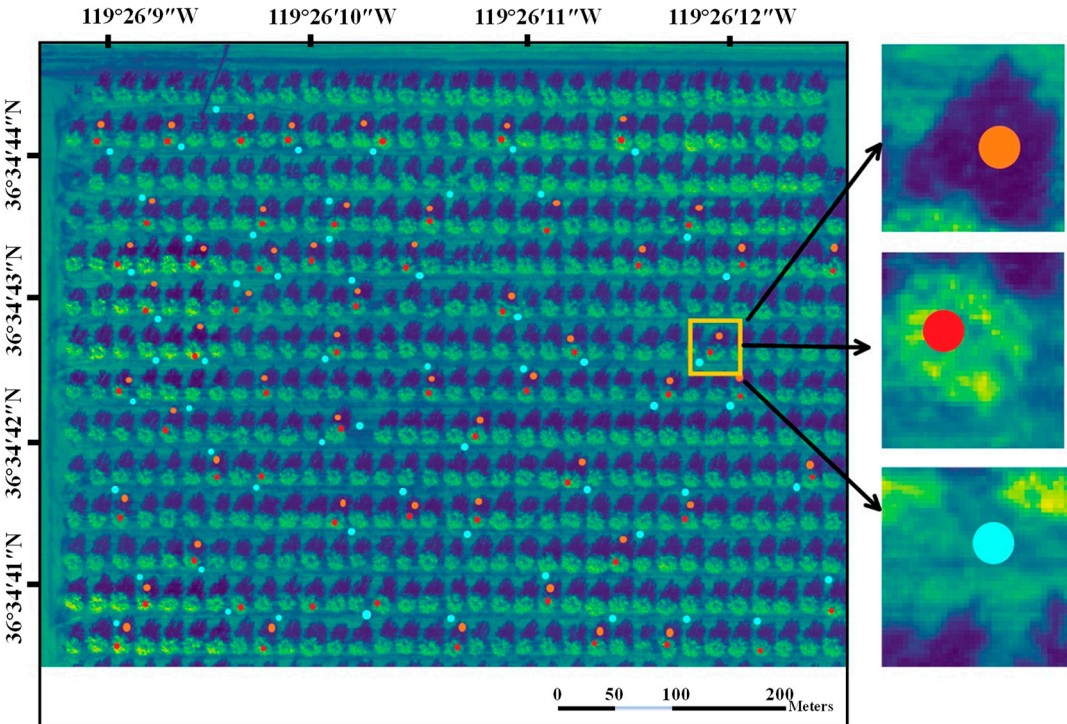

**Figure 3.** Randomly selected points for reflectance sampling. Red circles indicate trees, orange circles indicate tree shadows, and cyan circles indicate soil.

### 2.4. Performance Evalution

A variety of machine learning methods were employed here, including SVM, KNN, LDA, to investigate all possible models for classification fully. Some machine learning algorithms, like random forest and genetic algorithm, which have been used in a lot of machine learning researches, were not discussed in the fusion section. This was because these methods may have inconsistent results every time the training process was done. From this angle, they were not suitable for the goals to cross evaluate different machine learning methods. It is desirable to use methods that are consistent in performance to explore the feasibility of fusing vegetation indexes into a classification model.

To evaluate the model performance, we compared the output array with the test array. It was easy to tell how it performed by seeing that the output array from the predictive function was almost the same as the test array. In order to see the generalization of the model, the trained models were not only tested on the test set data but also applied to map the classification result of the entire research area. The trained model was used for pixel-wise classification, and the classification results are displayed in different colors. This was done to demonstrate the performance of the model. To visualize the classification result may not show too many details like the confuse matrix or outliers due to lack of ground truth data.



*2.5. Vegetation Index*

In this study, 11 VIs were used to improve classification accuracy. As the most prevailing VI, the sample radio (SR) is around 1, and SR is usually used for soil and vegetation segmentation since bare soils generally were near 1 and the number of green vegetation increased in a pixel (picture element), the SR increased. In other words, SR may contribute to the classification of the ground and the tree.

In the NDVI, the difference between the near-infrared and red reflectances is divided by their sum. Data from vegetated areas will yield positive values for the NDVI due to high near-infrared and low red or visible reflectances. As the amount of green vegetation cover increases in pixels, NDVI increases in value up to nearly 1. In contrast, bare soil and rocks generally show similar reflectance in the near-infrared and red or visible, generating positive but lower NDVI values close to 0. The red or visible reflectance of clouds, shadows are larger than their near-infrared reflectance, so scenes containing these materials produce negative NDVIs.

As shown in Table 2, all 11 types of VIs used in this study are listed. Before the data was used in further process, it needs to be normalized to the same scale. Their mutual relevances were studied, as Figure 4 shows. The closer the correlation coefficient was to 1 or −1, the more positively or negatively relevant they were and the darker the color got. The negative correlation and positive correlation were both taken into consideration. Dat and NDVI showed a high correlation $R^2 = 1$. SR and NDVRG were also the same. In general, their relevance was not very high. Nearly three-quarters of the vegetation index had a correlation coefficient below 0.5. We also hoped that the vegetation index correlation coefficient was lower so that different vegetation indices can play different roles.

**Table 2.** Vegetation indices used in this study, $\rho$ is the reflectance of band i.

| Index | Formula | Reference |
|:---:|:---:|:---:|
| Greenness Index (GR) | $\frac{\rho_{green}}{\rho_{red}}$ | Le Maire et al. (2004) [29] |
| Simple ratio (SR) | $\frac{\rho_{NIR}}{\rho_{green}}$ | Le Maire et al. (2004) [29] |
| Simple ratio Pigment specific SR B1 (SRB) | $\frac{\rho_{NIR}}{\rho_{red}}$ | Blackburn (1998) [30] |
| Datt2 (Dat) | $\frac{\rho_{NIR}}{\rho_{edge}}$ | Datt (1999) [31] |
| NDVI | $\frac{\rho_{NIR}-\rho_{red}}{\rho_{NIR}+\rho_{red}}$ | Tucker (1979) [32] |
| Green NDVI (NDVIG) | $\frac{\rho_{NIR}-\rho_{green}}{\rho_{NIR}+\rho_{green}}$ | Gitelson and Merzlyak (1997) [33] |
| Red-edge index (NDRE) | $\frac{\rho_{NIR}-\rho_{edge}}{\rho_{NIR}+\rho_{edge}}$ | Barnes et al. (2000) [34] |
| Red-edgeGreen NDVI (NDEGE) | $\frac{\rho_{edge}-\rho_{green}}{\rho_{edge}+\rho_{green}}$ | Buschmann and Nagel (1993) [35] |
| Red-edge NDVI (NDVGE) | $\frac{\rho_{edge}-\rho_{red}}{\rho_{edge}+\rho_{red}}$ | Ortiz et al. (2013) [36] |
| Anthocyanin reflectance index 2 (ARI2) | $NIR * \left[ \frac{1}{\rho_{green}} - \frac{1}{\rho_{edge}} \right]$ | Gitelson et al. (2002) [37] |
| EVI | $2.5 * \left[ \frac{\rho_{NIR}-\rho_{red}}{\rho_{NIR}+1*\rho_{red}-1*\rho_{blue}+1} \right]$ | Huete et al. (2002) [38] |

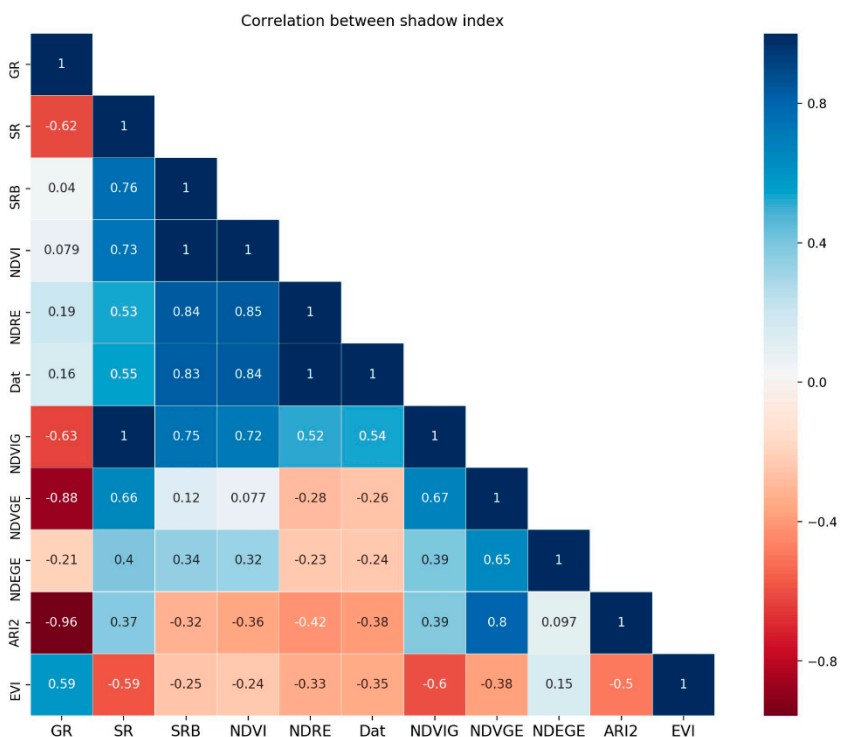

**Figure 4.** The VI correlation coefficient of different vegetation indices.

## 2.6. Supervised Learning Methods

Different supervised learning methods were implemented using the open-source Python Scikit-Learn toolkit. Support vector machine (SVM) and Kernel SVM are supervised machine learning classification algorithms. SVMs were introduced initially in the 1960s and were later refined in the 1990s. After 20 years of development, they are becoming extremely popular, owing to their ability to achieve outstanding results.

The training set contains five features, so it is very important to choose the appropriate kernel function. Various kernel functions were implemented, such as linear, nonlinear, polynomial, Gaussian kernel, Radial basis function (RBF), sigmoid. The trained model was used to predict all the raw data directly, and finally adopt linear as SVM kernel. The linear kernel function can effectively distinguish between trees and tree shadows, while other kernel functions work poorly.

K-NN is a type of instance-based learning, or lazy learning, where the function is only approximated locally, and all computation is deferred until classification. A peculiarity of the K-NN algorithm is that it is sensitive to the local structure of the data. The best choice of k depends upon the data; generally, larger values of k reduces the effect of the noise on the classification, but make boundaries between classes less distinct. The special case where the type is predicted to be the class of the closest training sample is called the nearest neighbor algorithm. We will set k to 1.

Linear Discriminant Analysis (LDA) is most commonly used as a dimensionality reduction technique in the pre-processing step for pattern-classification and machine learning applications. It has been used as a linear classifier to project a feature space (an n-dimensional dataset sample) onto a smaller subspace *k* while maintaining the biggest class-discriminatory direction.

Decision tree learning is a method commonly used in data mining. The goal is to create a model that predicts the value of a target variable based on several input variables. Random forests or random decision forests are an ensemble learning method for classification, regression, and other tasks that operate by constructing a multitude of decision trees at training time and outputting the class that is the mode of the types (classification) or

mean prediction (regression) of the individual trees. Random decision forests correct for decision trees' habit of overfitting to their training set

The idea of gradient boosting originated in Leo Breiman's observation that boosting can be interpreted as an optimization algorithm on a suitable cost function. Jerome H. Friedman subsequently developed explicit regression gradient boosting algorithms.

The model was trained firstly with the training set. To assess the performance of machine learning models, cross validation was used. This helps to know the machine learning model would generalize to an independent data set. K-fold cross validation is a common type of cross validation. As shown in Figure 5, firstly, the original training data set was partitioned into 6 equal subsets. Each subset was called a fold. Let the folds be named as f1, f2, . . . , f6. Secondly, keep the fold fi (i = 0, . . . , 6) as validation fold and keep all the remaining 5 folds in the cross validation training set. Lastly, we will estimate the accuracy of the machine learning model by averaging the accuracies derived in all the k cases of cross validation, and it is expressed by the symbol E. mean-square error can measure the average of the squares of the errors. MSE will also be an important indicator of the evaluation model.

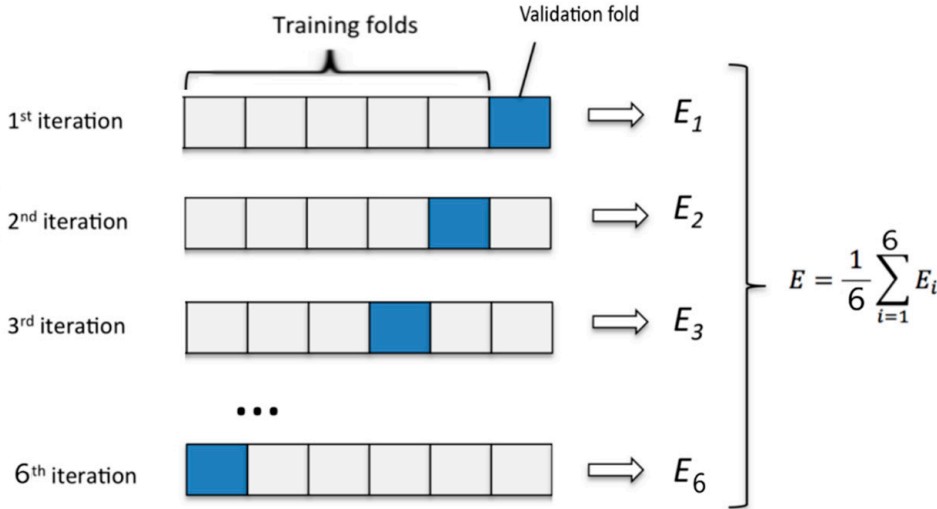

**Figure 5.** Distribution of training and test folds during cross validation.

Cross validation results are shown in Table 3, MSE is the mean squared error. In the supervised learning method, the E value of LDA was higher, and the MSE value was lower. SVM and KNN models were acceptable. The remaining supervised learning methods had lower E values, and MSE was not ideal. In the following research, we will focus on SVM, KNN, and LDA. The table shows the mean square error and the average correct rate under different supervised learning methods.

**Table 3.** The recognition accuracy of cross validation.

| Supervised Learning | Error (E) | Mean Squared Error (MSE) |
|---|---|---|
| SVM | 0.9312 | 0.000299 |
| KNN | 0.9187 | 0.000247 |
| LDA | 0.9395 | 0.000282 |
| DecisionTree | 0.7708 | 0.001122 |
| RandomForest | 0.7875 | 0.002656 |
| GradientBoosting | 0.8104 | 0.002990 |

*2.7. VIs' Normalization for Fusion Study*

Test sets were used to predict SVM model accuracy, which was up to 93.27%, and the model was quite good. In order to improve the accuracy of the model, the VI referenced

above was used to expand the dimension of the training set. We regarded the original training set as a reference object and compared it with the training set with the VI.

Because the SVM was sensitive to the data's size, it was necessary to find a normalization method to put the data in the same range. In this research, we proposed a new formula to normalize the data. The formula (2) and (3) was as follows:

$$Q = \{x_1, x_2 \dots x_i\} \tag{2}$$

$$y = (2^n - 1)\frac{x_i - Min(Q)}{Max(Q) - Min(Q)} \quad n : \text{the bit depth of an image; } x_i : \text{VI value} \tag{3}$$

This formula can extend the VI to the same order of magnitude. Then the VI and other band set from a training data set were combined to train the model.

## 3. Results and Discussion

### 3.1. The Performance of A Single VI

As shown in Figure 6, Figure 6a shows the performance of the five-band training model. It will be treated as a reference in the following sections. Others show model validation results for five bands and different vegetation indices as a data set. Results showed that VIs influenced the accuracy of the classification model, and the model based on extended features, including Red Edge NDVI, RedEdge Green NDVI, and NDVI, was better trained than the original multispectral data set. By comparing (e), (i), and (j), the total correct rate of them was over the 5-band model.

In general, the recognition rate of the ground was high, while the recognition rate of the tree was not that high. It can be seen from the results shown in the Figure 7c, tree recognition rate could be improved by bringing in a suitable VI.

Meanwhile, it can be seen that some VIs could increase the recognition rate of shadows. Although the recognition rate of trees and shadows increased, the degree of the increase was not very large, and there was still a significant gap in the recognition rate.

As shown in Table 4, KNN and LDA model accuracy was obtained from the test set. VI could improve the accuracy of the KNN model; however, VI could not enhance the accuracy of the LDA model. Most of the VIs that could improve the SVM and KNN were the same. NDVI, NDVGE, and NDEGE were all conducive to the improvement of SVM and KNN model accuracy. NDVIG alone affected KNN model improvement. SRB did not affect the accuracy of the LDA model. Other VIs reduced the accuracy of the LDA model.

ARI2 greatly impacted the accuracy of the SVM model, just as EVI to the KNN model. However, the VI index had little effect on the accuracy of the LDA model. So, appropriate VI could increase the accuracy of the SVM and KNN models, but VI had little impact on the LDA model.

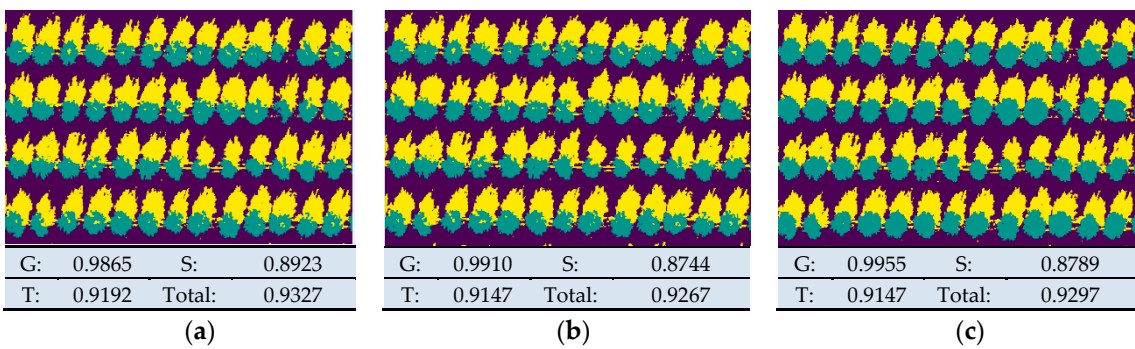

| G: | 0.9865 | S: | 0.8923 |
|---|---|---|---|
| T: | 0.9192 | Total: | 0.9327 |

**(a)**

| G: | 0.9910 | S: | 0.8744 |
|---|---|---|---|
| T: | 0.9147 | Total: | 0.9267 |

**(b)**

| G: | 0.9955 | S: | 0.8789 |
|---|---|---|---|
| T: | 0.9147 | Total: | 0.9297 |

**(c)**

**Figure 6.** *Cont.*

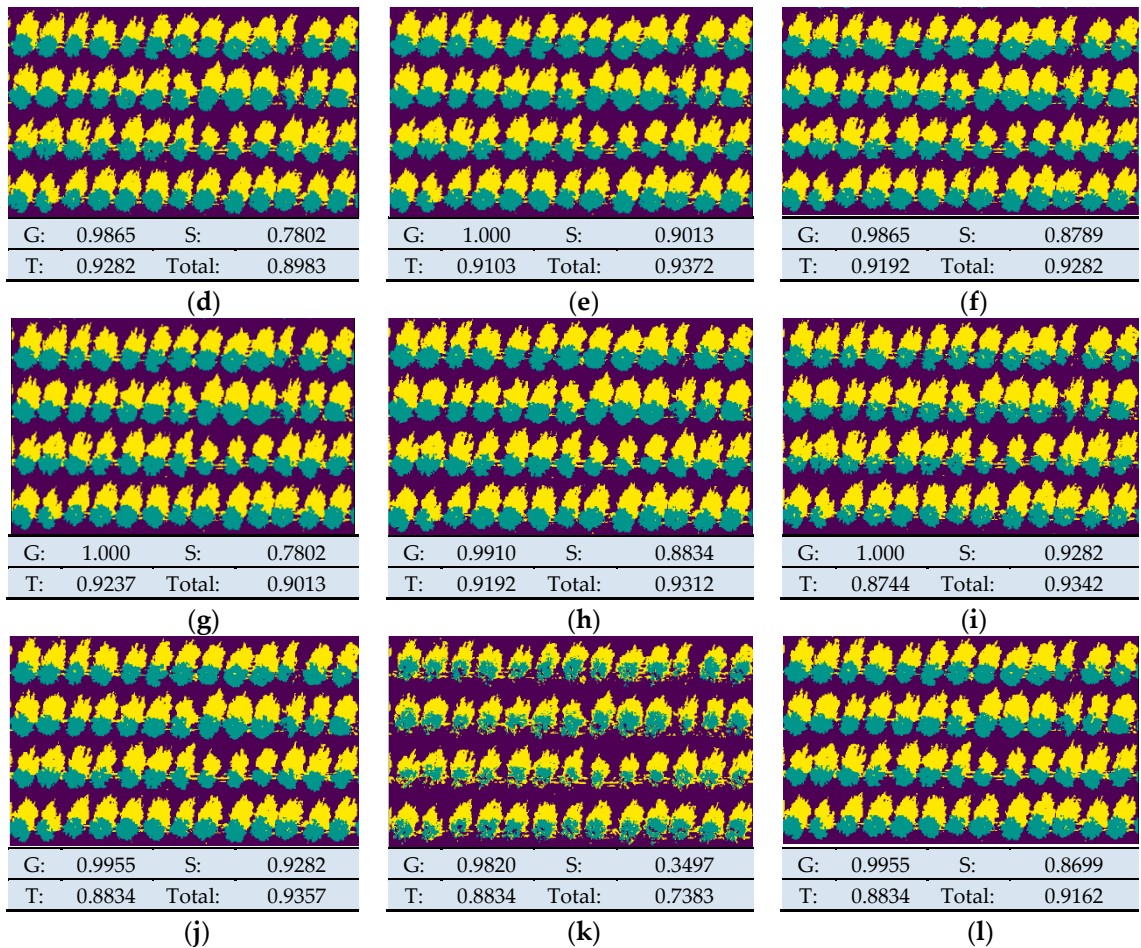

**Figure 6.** Schematic diagram of classification effect under different vegetation indices. G, T, and S represent the classification accuracy of ground, tree, and shadow, respectively, and T represents the total recognition accuracy. The image is classified using different VI indices. (**a**): Reference; (**b**): Green_ Index; (**c**): SR_ Index; (**d**): SRB_ Index; (**e**): NDVI_ Index; (**f**): NDRE_ Index; (**g**): Datt_ Index; (**h**): NDVI_ Green_ Index; (**i**): NDVI_ RedEdge_ Green_ Index; (**j**): NDVI_ RedEge_ Index; (**k**): ARl2_ Index; (**l**): EVI_ Index.

**Table 4.** The classification accuracy under different classification methods of other VIs, bold refers to those higher than the reference.

| VIs | SVM | KNN | LDA |
|---|---|---|---|
| reference | 0.9327 | 0.9267 | 0.9372 |
| GR | 0.9267 | 0.9267 | 0.9357 |
| SR | 0.9297 | 0.9058 | 0.9342 |
| SRB | 0.8983 | 0.8953 | **0.9372** |
| NDVI | **0.9372** | **0.9327** | 0.9342 |
| NDRE | 0.9282 | 0.9222 | 0.9327 |
| Dat | 0.9013 | 0.8938 | 0.9357 |
| NDVIG | 0.9312 | **0.9312** | 0.9357 |
| NDVGE | **0.9342** | **0.9417** | 0.9357 |
| NDEGE | **0.9357** | **0.9357** | 0.9282 |
| ARI2 | 0.7383 | 0.9103 | 0.9342 |
| EVI | 0.9162 | 0.7443 | 0.9207 |

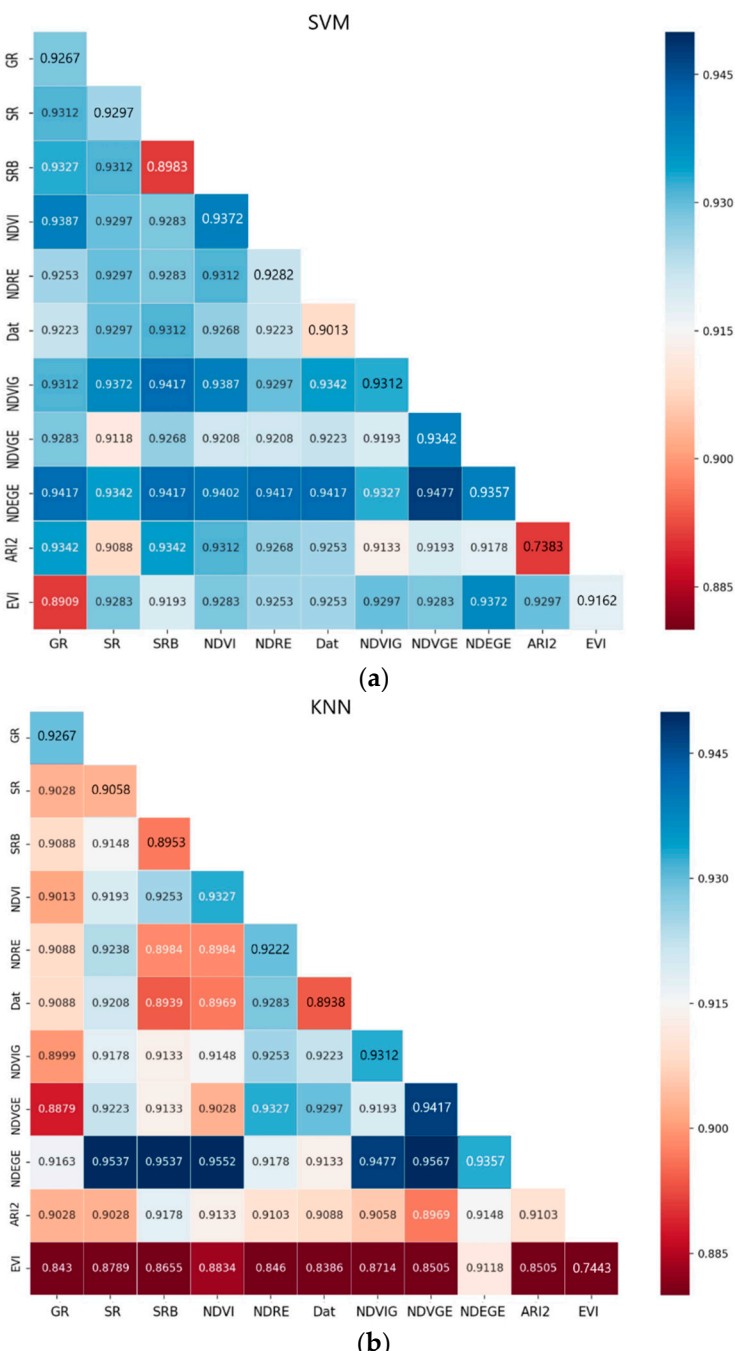

**Figure 7.** *Cont.*

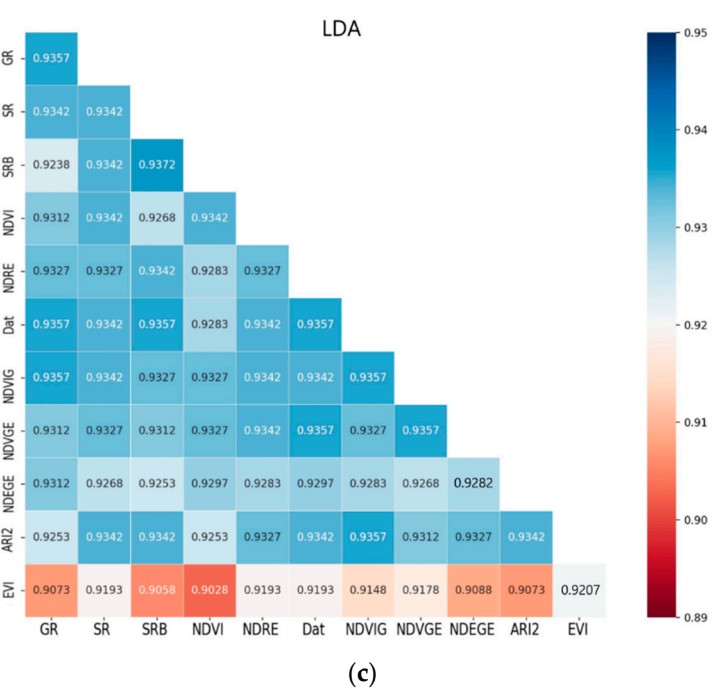

**Figure 7.** The correct recognition rate of the model under different VI indices, the deeper the blue degree, the higher the recognition accuracy, the deeper the redness, the lower the recognition rate. (**a**) Supervised learning method is SVM, (**b**) is KNN, (**c**) is LDA.

*3.2. The Performance of Two VIs*

To further explore the impact of the VI, we added two different vegetation indices to the feature space to see the classification results. The accuracy of the training model is shown in Figure 7a: The best SVM model recognition rate was up to 94.77%. Compared with the models with single VI added, the recognition accuracy improved a lot. The training data set using the single VI, the model trained by the NDVI_RedEdge_Green Vegetation index, was better. The dataset training rate of NDEGE VI and other index was relatively high. Compared with others, the model trained with the AIR2 was not as good as the dataset training rate of AIR2 and other indices were relatively low.

As shown in Figure 7b, the darker the blue was, the more accurate the model was. In general, the combination of the NDEGE index and other indices stood out from other hybrids. Research data showed that the model accuracy of NDEGE and NDVGE was the highest in the two-VI index results. Both NDVGE and NDEGE could improve model accuracy in the single index model. The two-VI model had a maximum accuracy of 95.67%, and it was more accurate than the single index model

In the KNN model, four indexes could improve the accuracy of the model. Their mutual combination training model did not necessarily improve the accuracy of the model. For example, NDVIG and NDVI had higher precision in their training models than other combinations. NDEGE worked well with training models with other VIs. For instance, NDEGE and these indices, which performed well in a single model, could improve model accuracy in the two-VI model. Both SR and SRB reduced the accuracy of the single VI model, but one of them combined with NDEGE to train the two-VI model with higher accuracy than the single VI model. It can be concluded that NDEGE not only benefited the accuracy of the model, but also combined with other VIs to improve the accuracy of the model.

As shown in Figure 7c, it can be concluded that the VI index did not improve the accuracy of the LDA model in the two-VI model. The impact of VI on the accuracy of the LDA model was also relatively small compared to the SVM model.

We observed the distribution of the VI index corresponding to the high-precision model. According to Figure 4, we can see that the correlation coefficient between these indices was not very high.

### 3.3. The Performance of Multiple VIs

To explore the influence of the input number of VIs on the accuracy of the model, different vegetation indices were extracted to form a new feature set to train the classification model. The result is shown in Figure 8a,c,e. The horizontal indicates the number of VIs used in the training set, and the vertical indicates the accuracy of the model. The scatter points in each vertical row are the models' accuracy under the same number of VIs. The plot was formed by the best model accuracy using different numbers of VIs.

When the number of VIs was up to three or four, the highest SVM model precision was 96.41%. Compared with the accuracy of the original five-band model, it improved by 3%. However, when using all VIs, accuracy was only 34.9%. When observing the single VI model, one of the indices could reduce the overall accuracy. When the index was combined with other indices to form a training set, many low-precision models were produced, which leads to lower performance. Figure 9 illustrates the outlier points in Figure 8a,c,e.

As shown in Figure 8b,d,f, the best accuracy of the model was initially increasing and then decreasing with the number of VIs. When the number of VIs were up to 3, KNN model achieved the highest precision. When the number of VIs was up to 3 or 4, SVM model achieved the highest precision.

As shown in Figure 8b,d,f, the horizontal indicates the number of VIs in the training set, and the vertical indicates all the VIs. These three graphs show the VI distribution of the best-trained model among all VI combinations. As it shows, VI fusion had mostly improved the accuracy of SVM and KNN models, but did not have much effect on the LDA trained model.

In the best SVM model, the NDEGE index appeared most frequently. In the single vegetation index SVM model, the NDEGE index could train the best model. NDEGE and NDVGE indices could train the best model in the two-VI SVM model. In three-VI SVM models, NDEGE, NDVGE, and NDVIG indices had the same effect as NDEGE, NDVGE, and NDVI indices. A combination of NDEGE, NDVGE, NDVIG, and NDVI indices could train the optimal model. It can be concluded that the NDEGE, NDVGE, NDVIG, and NDVI indexes were beneficial to improve the accuracy of the SVM model.

In the best KNN model, the NDVGE index appeared most frequently, In the single vegetation index KNN model, the NDVGE index could train the best model. NDVIG and NDVGE indices could train the best model in the two-VI SVM model. In three-VI KNN models, a combination of Dat, NDVGE, and NDVIG indices had the same effect as NDEGE, NDVGE, and NDVI indices, reaching an overall accuracy of 95.92%.

This study showed that a properly selected combination of Vis with original multispectral bands could improve the accuracy of the trained model, and a reasonable VI combination could further improve it.

To better understand the relationship between VIs and machine learning models. As Figure 9 shows, NDEGE and NDVGE were the two least occurred VIs among all combinations. The blank in each machine learning method was due to the lowest points and was not treated as an outlier.

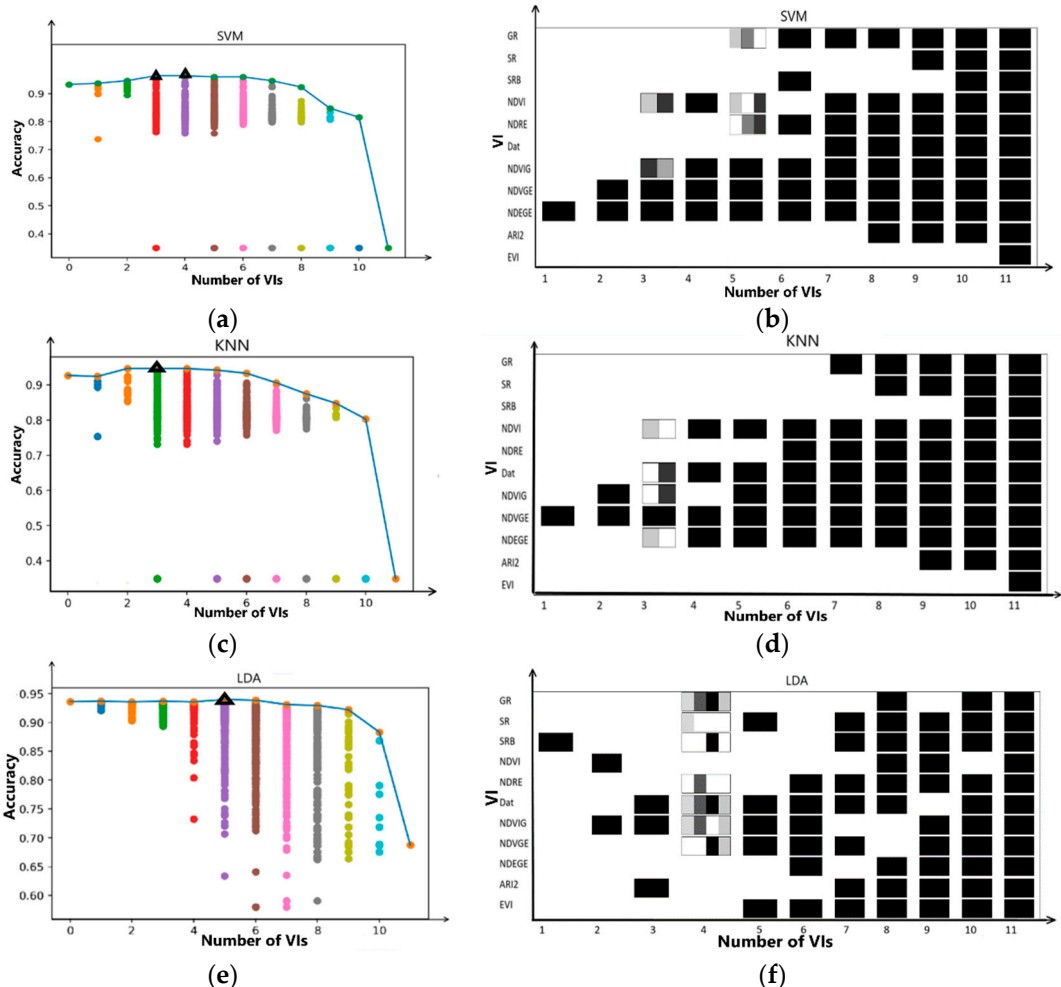

**Figure 8.** (**a**,**c**,**e**): The scatter points shows the distribution for the accuracy of different combinations. The curve consists of the highest precision under the same number of vegetation indices, and the black triangle indicates the highest point of the curve. (**b**,**d**,**f**): Show the distribution of vegetation indices under the best recognition accuracy. Squares of different gray levels are used to represent multiple combinations of different vegetation indices at the best accuracy, for example, 3 VIs in SVM of (**b**), black means this VI is shared by all combinations, white means not in any combination, light grey and dark grey mean this VI is in one variety to achieve the same best performance.

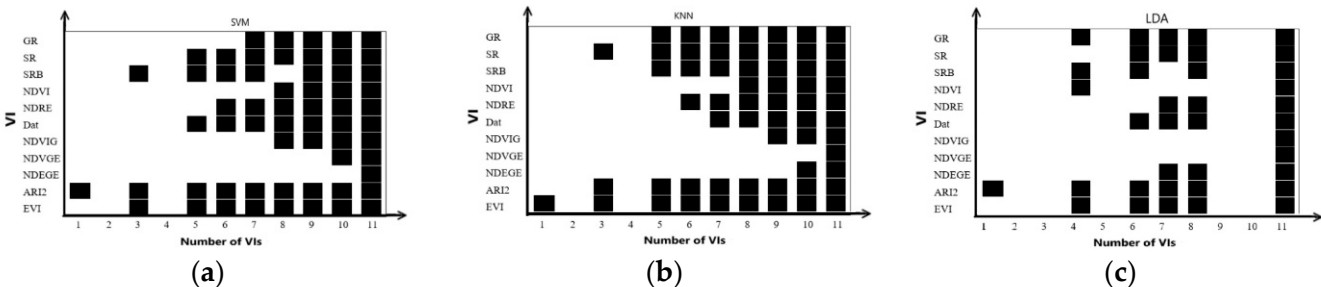

**Figure 9.** The combinations of worst performance.

### 3.4. Classification Accuracy

The trained model was applied to the entire orthoimage to demonstrate its classification accuracy. The set of original multispectral bands and added VIs were normalized as in the training process and put into the trained machine learning model. The best two-VI

SVM models were selected for this demonstration. The refractive indices of different classes in the same band are highlighted.

The best two-VI SVM models used for prediction were multi-spectrals combined with NDEGE and NDVGE. The segmentation result is shown in Figure 10. The classification effect showed it worked well, and the tree, shadow, and ground could be effectively distinguished.

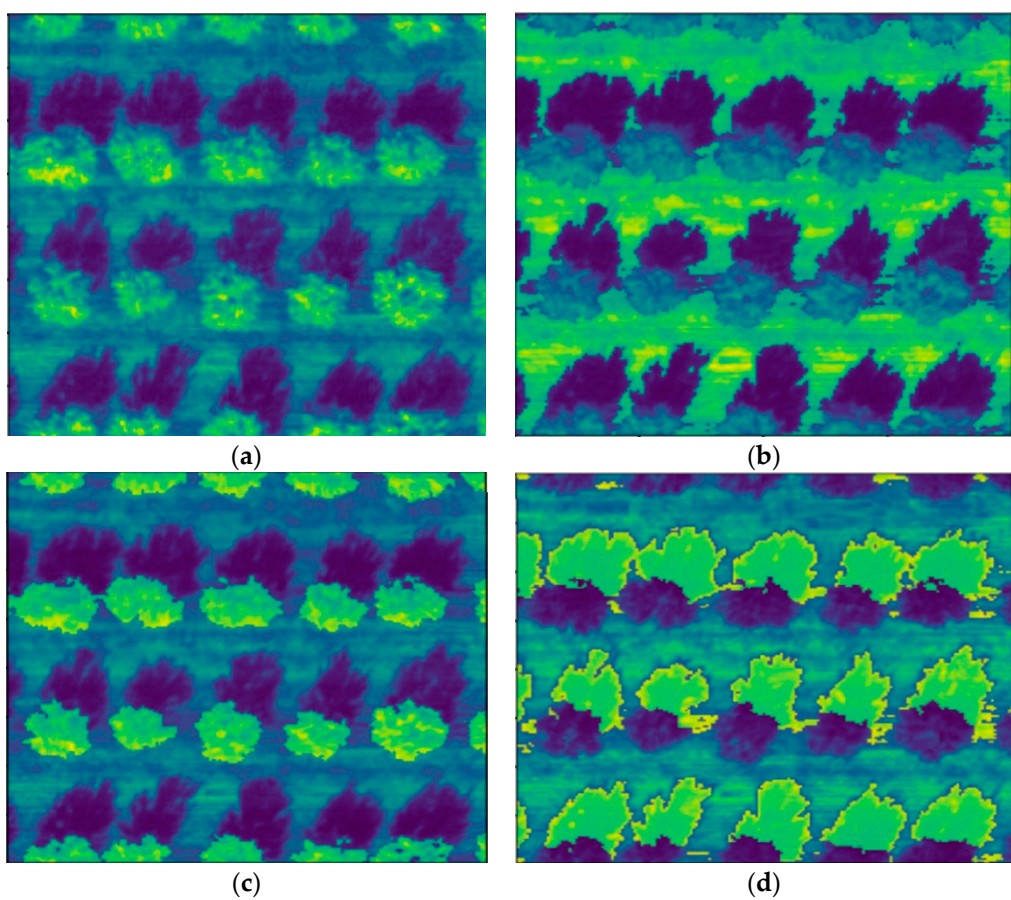

**Figure 10.** Visualization of Classification results based on different algorithm. (**a**) Reference; (**b**) ground; (**c**) tree; (**d**) shadow.

## 4. Discussions

Our results showed that the fusion of multispectral bands and VIs could improve the classification model accuracy if proper VIs were selected. In particular, three to four VIs could reach the optimal fusion result. All the results above could be attributed to three reasons to the best of our knowledge. First, the information in VIs could help enlarge the difference among ground covers. As discussed above, VIs were first introduced to describe the difference among ground covers so as to do tasks like classification, segmentation, monitoring. They contained nonlinear correlations between spectral bands that had not been introduced into machine learning-based remote sensing classification.

Second, different training methods were sensitive to different VIs. The selected training methods in this research were kernel-based SVM, distance-based KNN, linear transformation-based LDA. SVM, NDEGE, NDVIG, and NDVGE had a positive effect on building models. To KNN, the result was almost the same, NDEGE, NDVGE, and NDVI had a positive effect in improving model accuracy. To these two training methods, adding VIs created an apparent improvement. SVM was designed to find the hyperplane for class separation, so adding classes related features could improve the hyperplane performance. Meanwhile, the KNN separated types based on distance between new samples and known samples. KNN did not require training, but calculated distances every time, so it tended

to be very slow when the data set was large. From this point of view, adding VIs could increase the class distance to improve performance. One more thing was that all the VIs mentioned above were loosely correlated with each other, as is shown in Figure 4. For example, correlation coefficient between NDEGE and NDVIG was 0.39, NDVI and NDEGE was 0.32, NDVI and NDVGE was 0.077, NDVGE and NDEGE was 0.65. This may have been why the performance increased with more VIs, as Figure 8a,c shows.

However, LDA was different from the former two methods. LDA was based on linear transformation, and it was designed to project at the largest separation plane. So the performance was unlikely to change with the number increases because new information was unlikely to have influence on LDA projection. It can be concluded that LDA was not fit for information purposes.

Although combining spectral bands and VIs could improve the performance of the classification model, more was not better, and some VIs had a reverse effect. As it shows, the performance decreased when more than 4 VIs were added and dramatically dropped when all 11 VIs were added. Additionally, it can be seen there were points significantly lower than other points of the same column. We noticed that there were height-related indexes, such as CHM [39]. It could be used when the images were acquired at low altitude and the height was obvious in DSM. However, in our experiment, the altitude was 200 meters and the generated DSM had nearly no height information. Moreover, the resolution of multispectral camera was much lower than RGB cameras used in height-related studies. So, the DSM was flat and wasn't taken into further studies.

Moreover, the way to normalize VIs still needs more exploration. In this study, VIs were normalized to zero to one according to the minimum and maximum value of all pixels. This was because some VIs could not be normalized to a particular range. For example, SRB was the rate of NIR and Red, so it could be very big. There was hardly any other way to normalize it to a certain value range except the proposed method in this research. Meanwhile, this method could maximize the influence of other normalized VIs. For example, NDVI was an index already normalized between minus one and one, but in the normalized method used in this research, the value was normalized to the scale of minimum and maximum value of all pixels. In this way, the VI value was considered to be larger than it was.

## 5. Conclusions

This paper describes the application of a UAV-based five-band multispectral camera for the classification of three different ground covers, trees, soil, and shadow at first. To further utilize the nonlinear information between multispectral bands, a fusion method was developed to see if it can improve the performance of the classification model. The classification results based on the original five bands were 0.9327 for SVM, 0.9267 for KNN, and 0.9372 for LDA. It can be concluded that mixed ground cover like shadow can be properly distinguished. This result can be helpful in research using shadow-based measurement.

Secondly, 11 VIs were calculated and integrated with the original five bands to train models using three supervised learning algorithms, namely SVM, KNN, and LDA. After gradually adding VIs into feature space, it was found that three to four VIs could improve the classification performance when using SVM and KNN, and it decreased if more were added. VIs had little influence on LDA-based classification. It can be concluded that fusing multispectral bands and VIs in supervised learning can improve classification performance. Among all 11 proposed VIs, NDEGE, NDVIG, NDVGE, and NDVI positively affected information fusion.

At present, only VIs have been used and proved effective in remote sensing for information fusion in this research. The constructed VIs is limited due to limited bands of multispectral camera that can be mounted on a UAV. In this research, a five-band multispectral camera was used. In the future, with the development of spectral sensor technology, cameras with more bands can bring more VIs. Meanwhile, only several linear machine learning methods were cross-compared. This was to guarantee the consistency of

the trained model. In the future, far more complicated machine learning algorithms can be used to verify this research. At last, the classification result based on original bands was pretty good, and information fusion can improve the accuracy. It was more desirable to explore the effect of fusion in cases in which the classification result was bad based on the spectral bands' original information.

**Author Contributions:** Conceptualization, Y.Z.; Formal analysis, W.Y.; Funding acquisi-tion, Y.Z.; Methodology, W.Y.; Project administration, Y.Z.; Visualization, W.Y.; Writing—original draft, Y.Z., W.Y. and J.Y.; Writing—review & editing, Y.S., C.C. and W.Z. All authors have read and agreed to the published version of the manuscript.

**Funding:** This research was funded by the National Natural Science Foundation of China (NSFC), grant number: 61905219.

**Data Availability Statement:** The data source can be found at https://github.com/micasense/imageprocessing/tree/master/data (accessed on 30 March 2021).

**Acknowledgments:** We would like to thank Micasens Inc for sharing dataset with us.

**Conflicts of Interest:** The authors declare no conflict of interest.

## Abbreviations

The following abbreviations are used in this manuscript:

| | |
|---|---|
| SVM | Support Vector Machine |
| KNN | K-Nearest Neighbor |
| LDA | Latent Dirichlet Allocation |
| E | Error |
| MSE | Mean Squared Error |
| GR | Greenness Index |
| SR | Simple Ratio |
| SRB | Simple Ratio Pigment Specific SR B1 |
| Dat | Datt2 |
| NDVI | The Normalized Difference Vegetation Index |
| NDVIG | Green NDVI |
| NDVIG | Red-edge index |
| NDVIG | Red-edge Green NDVI |
| NDVGE | Red-edge NDVI |
| ARI2 | Anthocyanin reflectance index 2 |
| EVI | The Enhanced Vegetation Index |
| DSM | Digital Shape Model |

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
