# Peer review of "Fusion of Multispectral Aerial Imagery and Vegetation Indices for Machine Learning-Based Ground Classification"

_remotesensing, doi:10.3390/rs13081411_

Round 1
Reviewer 1 Report
The paper appears to read well and to be well researched and structured. I would perhaps like to see more context surrounding the value of this research to application. At present you have UAV imagery, a method that provides the opportunity to extract more information from the imagery, but I wondered about the overall context of UAV image classification and information extraction? Perhaps some more background could also be provided. Minor things ... some editing of the English in the text... and an illustration of the drone / camera system.
Author Response
Your comments were highly insightful and enabled us to improve the quality of our manuscript. The overall context of UAV image classification can be divided into two main categories. One is the spectral information-based classification. For example, RGB, NIR, see ref [1,2], and the classification and segmentation based on proposed vegetation indexes[3]. The other is texture/contour/shape-based classification, For example, texture classification[4]deep learning, see ref [5]. We agree with you that more background could be provided and we made changes in the Introduction section. Please see the revised version.
We have made corrections to the language and added an illustration of the drone/camera system in the figure. Please find them in the revised version.
[1] Zhang M, Zhou J, Sudduth K A, et al. Estimation of maize yield and effects of variable-rate nitrogen application using UAV-based RGB imagery[J]. Biosystems Engineering, 2020, 189: 24-35.
[2] Marcial-Pablo M J, Gonzalez-Sanchez A, Jimenez-Jimenez S I, et al. Estimation of vegetation fraction using RGB and multispectral images from UAV[J]. International journal of remote sensing, 2019, 40(2): 420-438.
[3] Hassan M A, Yang M, Rasheed A, et al. A rapid monitoring of NDVI across the wheat growth cycle for grain yield prediction using a multi-spectral UAV platform[J]. Plant science, 2019, 282: 95-103.
[4] Kwak G H, Park N W. Impact of texture information on crop classification with machine learning and UAV images[J]. Applied Sciences, 2019, 9(4): 643.
[5] Chen Y, Lee W S, Gan H, et al. Strawberry yield prediction based on a deep neural network using high-resolution aerial orthoimages[J]. Remote Sensing, 2019, 11(13): 1584.
Reviewer 2 Report
Abstract: the selected VI's are not so faniliar. Please write also the whole names.
ch.1: The connection of 9th citated article to this article is questionable.
21 th citation does not fit the topic; other resolution, other context.
line 80: why blue letters?
ch. 2.1.: row spacing means row spacing of trees? Or what?
line 123. and 125: why red letters?
Fig.2. You can mention that Vegetation index is the fig.4. and there it is readable. - Do you use VI's in the supervised learning pf all ML methods or just at the selected ML methods (after cross-validation)? -Please citate this fig. in the text. -I do not understand dotted lines and VI1s improve your accuracy that is why the line at the right side should have the array two-wayed.
Fig.3. Please add scale to map.
line 161.: ratio (not radio) and ...to 1,... (not 'to1,'
Title of table 2. ...where is the 'p' the reflectance...
You did not add the indexes's literature of Table 2. to references!? And some literature are not the original article of the index (fi. NDVI and EVI)
line 200. The phrase has not end.
Paragraph of lines between 210-218: big letters in the middle of the phrases.
Table 3. Bad edit in the whole title
Eq.2. {x1, x2... (not 'x1, x1,...')
line 243: 'cite your figures' ? And next paragraph (from the line 244) The values of 'c' do not show this statement. Mayve 'd'?
Fig.6. Why do not you write to the little figs the name of the VI's and not only in the title. Why is the title of Fig.6. in another page.
Line 257. greatly impact is in real a 'negative' impact?
Table 4. You should mention in the title why are bold leters here?
Fig 7. I don't understand clearly why do not have the value 1 of the same place like fig.4.?
Ref. 28 has a wrong bibliography at Ref. list.
Conclusion: Can you see when it might be utilised in real conditions and not just in special case like now?
Author Response
Your comments were highly insightful and enabled us to greatly improve the quality of our manuscript. The attached word file is our point-by-point response to each of the comments
